# Is Brazilian Jiu-Jitsu a Traumatic Sport? Survey on Italian Athletes’ Rehabilitation and Return to Sport

**DOI:** 10.3390/jfmk10030286

**Published:** 2025-07-25

**Authors:** Fabio Santacaterina, Christian Tamantini, Giuseppe Camarro, Sandra Miccinilli, Federica Bressi, Loredana Zollo, Silvia Sterzi, Marco Bravi

**Affiliations:** 1Rehabilitation Unit, Fondazione Policlinico Universitario Campus Bio-Medico, Via Alvaro del Portillo, 200, 00128 Rome, Italy; f.santacaterina@policlinicocampus.it (F.S.); giuseppe.camarro@gmail.com (G.C.); s.miccinilli@policlinicocampus.it (S.M.); f.bressi@policlinicocampus.it (F.B.); s.sterzi@policlinicocampus.it (S.S.); m.bravi@policlinicocampus.it (M.B.); 2Research Unit of Advanced Robotics and Human-Centred Technologies, Department of Engineering, Università Campus Bio-Medico di Roma, Via Alvaro del Portillo, 21, 00128 Rome, Italy; l.zollo@unicampus.it; 3Institute of Cognitive Sciences and Technologies, National Research Council of Italy, Via Giandomenico Romagnosi 18a, 00196 Rome, Italy; 4Department of Medicine and Surgery, Università Campus Bio-Medico di Roma, Via Alvaro del Portillo, 21, 00128 Rome, Italy; 5Fondazione Policlinico Universitario Campus Bio-Medico, Via Alvaro del Portillo, 200, 00128 Rome, Italy

**Keywords:** Brazilian Jiu-Jitsu, injury prevention, psychological readiness, combat sports

## Abstract

**Background:** Brazilian Jiu-Jitsu (BJJ) is a physically demanding sport associated with a notable risk of musculoskeletal injuries. Understanding injury patterns, rehabilitation approaches, and psychological readiness to return to sport (RTS) is essential for prevention and management strategies. This study aimed to investigate injury characteristics among Italian BJJ athletes, assess their rehabilitation processes and psychological recovery, and identify key risk factors such as belt level, body mass index (BMI), and training load. **Methods:** A cross-sectional survey was conducted among members of the Italian BJJ community, including amateur and competitive athletes. A total of 360 participants completed a 36-item online questionnaire. Data collected included injury history, rehabilitation strategies, RTS timelines, and responses to the Injury-Psychological Readiness to Return to Sport (I-PRRS) scale. A Random Forest machine learning algorithm was used to identify and rank potential injury risk factors. **Results:** Of the 360 respondents, 331 (92%) reported at least one injury, predominantly occurring during training sessions. The knee was the most frequently injured joint, and the action “attempting to pass guard” was the most reported mechanism. Most athletes (65%) returned to training within one month. BMI and age emerged as the most significant predictors of injury risk. Psychological readiness scores indicated moderate confidence, with the lowest levels associated with playing without pain. **Conclusions:** Injuries in BJJ are common, particularly affecting the knee. Psychological readiness, especially confidence in training without pain, plays a critical role in RTS outcomes. Machine learning models may aid in identifying individual risk factors and guiding injury prevention strategies.

## 1. Introduction

Practicing martial arts offers numerous advantages, including the development of discipline, respect, strength, coordination, balance, and flexibility [1,2].

Among the various martial arts, Brazilian Jiu-Jitsu (BJJ) stands out due to its emphasis on throws, joint locks, and strategies to immobilize opponents, allowing individuals of smaller stature or weaker physical strength to successfully subdue larger or stronger opponents using proper technique. However, the literature lacks clarity on the risks associated with regular training [3].

Currently, in Italy, the number of BJJ athletes and interest in this sport is steadily increasing. The Italian Federation of Judo, Wrestling, and Karate Martial Arts (FIJLKAM), affiliated with the Italian National Olympic Committee (CONI) has integrated BJJ among its federal disciplines, delegating management to the Italian Jiu Jitsu Union (UIJJ). This shift follows years during which the Italian Federation of Grappling and Mixed Martial Arts (FIGMMA), an autonomous organization, managed BJJ and Mixed Martial Arts (MMA) sectors. This surge in sports interest, however, is not matched by equivalent scientific interest. Compared to similar sports like Judo [4] or freestyle wrestling [5], there are limited data and studies available on BJJ.

In any sport, injuries can occur despite precautions. Although the literature highlights a high injury rate in combat sports like Judo [6], Shotokan Karate [7], Boxing [8], Wrestling [9], epidemiological data, injury types, mechanisms, and especially injury management and return to sport (RTS) in BJJ are still limited.

The available literature indicates that the most affected body segments are distal, with fingers and toes as the structures reporting the most injuries, often undiagnosed or untreated medically [3,10]. The elbow and knee [11,12,13] are the joints most prone to injuries requiring medical consultation.

Some studies suggest a lower injury risk in athletes with lower belt levels (white, blue) compared to higher belt levels (purple, brown, and black), though the data is not always significant [11]. Additionally, there is disagreement regarding the injury setting between less and more experienced athletes. According to some authors [3,14], less experienced athletes are more likely to get injured during training rather than in competitions. Other studies dispute the validity of this finding [10,13].

Due to this heterogeneity in results, it is crucial to better explore these topics with studies providing clearer insights.

The primary goal of this investigation was to identify the types, locations, and most frequent traumatic mechanisms of injuries, as well as commonly employed treatment and rehabilitation methods. The secondary objective was to explore potential risk factors correlated with more severe or frequent injuries, such as age, participation in competitions, athlete experience, training methods, and sports habits. Additionally, this study examined the methods and decision-making processes for RTS.

In contrast to the existing literature, this study not only conducted an epidemiological analysis of sports injuries in BJJ but also examined the injury management methods chosen by athletes, the role of physiotherapy in this process, and whether RTS considered the athlete’s Psychological Readiness, assessed through the administration of I-PRRS [15].

The hypothesis of this study, which explores the Italian BJJ sports landscape, was that the frequency of injuries is comparable to that observed in other countries, with the knee as the most injured joint and that a correlation exists between injuries and participants’ belt levels, BMI, and training loads.

## 2. Materials and Methods

### 2.1. Study Design

An observational cross-sectional study was conducted in accordance with the Strengthening the Reporting of Observational Studies in Epidemiology (STROBE) guidelines (Appendix A) [16]. To gather data, a 36-item online questionnaire was created using Microsoft Forms and targeted both Italian and non-Italian BJJ athletes living in Italy. The purpose of the questionnaire was to explore athletes’ training habits, how they managed their most severe injuries, their experiences with RTS, and the psychological and athletic effects of those injuries. The project was registered on the Open Science Framework (OSF) in August 2023 under the identification code 10.17605/OSF.IO/2BTFR. Ethical approval for this study was granted by the Campus Bio-Medico University Ethical Committee on 31 May 2023, with protocol number PAR 83.23 OSS.

### 2.2. Questionnaire

A 36-item questionnaire was developed, building on the one designed by Petrisor et al. [17]. Additional questions were incorporated to more comprehensively analyze injury management, the decision-making criteria for RTS, and the impact of injuries on both the athlete’s sporting and non-sporting routines. To further assess Psychological Readiness, six items from the I-PRRS were also included [15]. The questionnaire underwent pretesting with three BJJ athletes, three coaches, and three physiotherapists with expertise in combat sports. This process aimed to assess face and content validity, ensuring that the items were grammatically correct, logically consistent in Italian, and to ensure that all pertinent aspects of sports practice, including trauma and its management, were adequately addressed [18]. While this step did not constitute a full psychometric validation, it allowed for refinement of the questionnaire to align with the cultural and sport-specific context of the Italian BJJ community. Following this review, the questionnaire was distributed from May 2023 to November 2023 to instructors and athletes from various BJJ gyms and dojos, both nationally and internationally. All participants confirmed that they had reviewed the informed consent form and agreed to the processing of their data for research purposes.

The first section of the questionnaire, comprising 15 questions, aimed to investigate general information about the athlete, such as gender, age, weight, height, profession, and health status. Additionally, it explored the athlete’s sporting achievements, including the achieved belt rank, participation in competitions in the year prior to completing the questionnaire, training frequency and duration each week, and involvement in complementary sports activities alongside BJJ.

The second section included 15 questions concerning the injuries that each athlete had encountered in the 3 years preceding the questionnaire’s completion, both during competitions and training sessions. To focus this study on the most significant cases, we explored their most severe injury, specifically the one that caused the greatest concern, resulted in the longest suspension from sports activity, or induced the most pain or functional limitations.

This investigation covered the injury’s location, type, traumatic mechanism, the context in which the injury occurred, the duration of the suspension from sports activity, the injury management methods, and the consequences produced by the injury in terms of sporting habits and performance.

The goal was to dig into the relationships between the injury, the athlete’s management strategies, and their physical characteristics to understand whether and how these variables impacted the RTS activity, both positively and negatively.

In the final section of the questionnaire, the athlete’s psychological profile was assessed using the 6 items of the I-PRRS scale, developed by Glazer in 2009 [19] and validated in Italian by Conti and colleagues a decade later [15]. The scale investigates two areas:•Confidence in performance capability.•Confidence in recovery.

Despite a significant increase in research on “Psychological readiness” for RTS in recent years, there remains a conceptual lack of clarity on what this concept entails [20]. Currently, there is a wide variety of approaches to define the concept of Psychological Readiness and its components in the literature. However, “psychological readiness” is generally considered from the perspective of willingness to act [21].

Moreover, physical parameters are often the sole data considered to evaluate an athlete’s form and their RTS. In contrast to this trend, several studies have suggested that athletes should also be psychologically prepared to return to performing safely and efficiently [22]. Additionally, various researchers have emphasized that the achievement of physical and psychological readiness may not always coincide [23], underscoring the importance of an assessment in both domains.

Qualitative examinations of psychological characteristics during the post-injury phases have highlighted various psychological factors, such as mood, that can influence the return to pre-injury levels both positively and negatively [15].

“Athletes may be considered psychologically ready when they possess psychological resources that facilitate a safe, productive, and enjoyable return to sports” [21]. Confidence is undoubtedly a key component of psychological readiness [15,21,24], encompassing not only confidence in the recovery of injured areas but also confidence in the ability to perform activities successfully [19,23].

### 2.3. Enrolled Participants

All BJJ athletes of Italian nationality who declared that they carry out sporting activities on Italian territory were included in this survey. To this end, several gyms located throughout Italy and the most important competitive events of 2023 in Italy were used to disseminate the questionnaire.

The inclusion criteria were as follows:•Professional or amateur BJJ athlete aged 18 years or older.•Adequate understanding of the Italian language.•BJJ athlete training in Italy or an Italian BJJ athlete training abroad.•Agreement to provide informed consent.

The exclusion criteria were as follows: •Incomplete questionnaire.•Failure to provide informed consent.

### 2.4. Study Size

The sample size was calculated using the formula proposed by Taherdoost [25] and used in other surveys [26,27]. The reference population considered comprised the athletes affiliated with BJJ Italy as of 16 March 2023 (n = 5431). To achieve statistical validity, it was necessary to reach a sample size of at least 357 responses. This approach would yield a margin of error of 5% and a confidence level of 95% [25].

### 2.5. Variables

The main objective of this study was to investigate the primary characteristics and dynamics of injuries in BJJ. Given the absence of a standardized questionnaire for this purpose, one was created to collect general information from each individual, details about their sports activity, and information on previous injuries, with a particular focus on their most severe injury. The questionnaire delved into the management of such injuries and their long-term consequences.

### 2.6. Analysis

All the results were analyzed using descriptive statistics to provide data on the frequency (counts, percentages, means, and standard deviations) of the respective responses. Microsoft Excel 365 v.2211 (Microsoft Corporation, Redmond, WA, USA) was used for the analysis.

Subsequently, a feature ranking analysis was carried out to determine the factors contributing to injury occurrence among BJJ athletes. More in detail, the Random Forest algorithm, known for its robustness in handling complex datasets, was chosen for its ability to rank the importance of various features. Random Forest builds multiple decision trees and aggregates their predictions, providing a reliable feature importance ranking while mitigating the risk of overfitting. For the feature ranking analysis, we developed three models: •Training Injuries: The first model aimed to classify the presence of injuries sustained during training sessions. The dataset was split into training and test sets using stratified sampling, with 30% of the data allocated to the test set to maintain the distribution of the target variable.•Competition Injuries: The second model was designed to classify injuries sustained during competitions, following the same procedure of stratified sampling and training-test split.•Overall Injuries: The third model combined training and competition injuries to assess the overall injury risk.

In each case, feature importance was computed based on the Random Forest model’s ability to correctly classify the injury outcomes. This information was subsequently used to identify the most influential variables for further analysis.

Moreover, we focused on the three most important variables identified in the Random Forest models. These variables were subjected to a deeper examination to understand their distributions across athletes with and without injuries. We used violin plots to visualize the distribution of these variables, stratifying the data by the presence of injuries. The analysis was further divided based on gender to explore potential differences between male and female athletes. To evaluate the statistical significance of the observed differences, we applied the Mann–Whitney U test, a non-parametric test suitable for comparing two independent groups, with a significance level of 0.05. The data analysis was conducted using Python 3.8, utilizing key libraries such as scikit-learn for machine learning algorithms, specifically Random Forest classification, pandas for data manipulation and preprocessing, matplotlib and seaborn for data visualization, and statannotations for the application of statistical tests and annotations to enhance the interpretability of the visual results.

## 3. Results

### 3.1. Participants

The survey was shared via multiple channels, including messaging apps and social media, gathering a total of 360 anonymous responses. All the 360 questionnaires collected were analyzed. The main demographic characteristics are summarized in Table 1, and the data related to the employment of the respondents is available in Table 2. The geographical distribution is represented in Figure 1. In total, 29.2%, which was the majority of those interviewed, declared that they had a blue belt (105), 27.8% (100) had a white belt, 18.6% (67) had a purple belt, 11.3% (41) had a brown belt, and 13.1% (47) had a black belt (Table 3). Table 4 shows the Gi and No-Gi frequencies and percentages during training.

### 3.2. Injury Characteristics and Management

Table 5 summarizes the data related to the number and percentage of athletes interviewed based on the belt rank achieved and the total number of injuries over three years during training and competitions, stratified by belt rank achieved.

Regarding injuries, 29 athletes stated that they had not suffered any injuries either in training or competition in the past 36 months. We collected 331 reported injuries; almost all occurred during training (n = 291; 88%), while the remaining happened during a competition (n = 40; 12%).

The traumatic mechanisms that most frequently led to injury were “while attempting to pass guard” (n = 86; 26%) (Figure 2). If we consider the traumatic mechanisms during training (n = 291), the most frequent were “while attempting to pass guard” (n = 79; 23.8%). Considering the injuries that occurred during competition (n = 40), the most frequent injury mechanism was ‘while being submitted’ (n = 18; 45%).

Although various types of injuries were reported (Figure 3), the majority of these involved the knee joint, accounting for 103 out of 331 reported injuries (31.1%), followed by the shoulder joint with 42 cases (12.7%) and the ankle joint with 27 cases (8.1%). This distribution is similar both among lower belt ranks (white, blue) and more experienced ranks (purple, brown, and black) (Figure 4).

The reported injuries were primarily sprains (n = 91; 27.5%), followed by strains (n = 38; 11.5%), fractures (n = 38; 11.5%), contusions (n = 35; 10.5%), and dislocations (n = 32; 9.6%). Of the athletes interviewed, 91 (27.5%) reported sustaining an injury that did not fall under the predefined categories and was therefore classified as ‘Other’.

Following their injuries, each athlete had to refrain from sports activities for varying periods of time (Table 6). Specifically, 217 athletes (65.5%) returned to practicing BJJ within a month of the traumatic event, 69 athletes (20.8%) returned within 3 months, 17 athletes (5.1%) within 6 months, 7 athletes (2.1%) within 12 months, while 3 athletes (<1%) returned after more than 12 months. Meanwhile, 18 athletes (5.4%) were still injured or had not yet returned to practicing the sport following the injury.

Of the athletes injured, 165 sought helps from a physiotherapist, 69 opted for medication, 57 went to the emergency room, 29 underwent surgery, while 98 did not require any type of treatment. The athletes could choose more than one answer for this question.

Among those who sought help from a physiotherapist, 77 included physical therapies in their treatment plan; 100 had therapeutic exercise programmed; 120 received manual therapy; and 47 required therapeutic education treatment. The athletes could choose more than one answer to this question.

### 3.3. Injuries, Consequences, and RTS

For the analysis of data regarding changes in sports habits and the RTS of athletes, only the responses of those who reported at least one injury, in training, or in competitions were considered (n = 331). The I-PRRS scale scores are summarized in Table 7.

### 3.4. Random Forest Algorithm

The Random Forest algorithm was applied to predict the likelihood of injuries during training, competition, and overall contexts. The performance of the classifier varied across the three cases. For training-related injuries, the model exhibited an accuracy of 87.04%, demonstrating strong predictive power in identifying factors that contribute to injuries during training sessions. The accuracy of the model decreased slightly when applied to competition-related injuries, reaching 74.07%. This indicates reasonable prediction accuracy, though lower than for training injuries, likely due to the distinct nature of competitive environments. When both training and competition injuries were combined, the model’s accuracy was 75%, reflecting a balanced performance when considering the overall context of injury occurrence.

The feature importance derived from the Random Forest models provided insight into the most influential variables driving injury risk. Figure 5 illustrates the ranking of feature importance across the three models, highlighting the most critical variables for injury prediction in BJJ athletes. As shown in Figure 5, BMI emerged as the most significant factor for predicting injuries, particularly during training, where it showed the highest importance value. This suggests that body composition may play a role in injury susceptibility during training sessions. Age also consistently showed high importance across all injury categories, underscoring its relevance in both training and competition-related injuries.

In terms of competition-related injuries, the number of competitions participated in (shown as # competitions in Figure 6) was particularly influential. This variable demonstrated the highest importance for predicting overall injuries, indicating a strong relationship between competition frequency and injury risk. Additionally, factors such as Experience and Gi vs. No-Gi were consistently relevant across the models, suggesting that both general experience and the type of training could influence injury risk.

Other variables, such as Training time and engagement in other sports, also contributed to the predictions, although their importance was more moderate. Variables related to Location, Profession, and Level showed lower importance, though they still provided some predictive value, particularly in the competition injury model.

The violin plots reported in Figure 6 provide the representation of the distribution of key variables across athletes, stratified by sex and injury status in the left column, with respect to the increasing number of reported injuries in the right column.

Athletes with a history of injuries, whether sustained during training or competition, display distinct distributions in terms of BMI, age, and number of competitions compared to non-injured athletes. Those with injuries generally have higher BMI values and are older, with injury prevalence increasing among older individuals. Additionally, athletes with injuries tend to have participated in a greater number of competitions.

When considering the number of injuries, the data reveal that athletes who have sustained more injuries tend to exhibit higher BMI and a higher frequency of competition participation, particularly among male athletes. This suggests a strong correlation between competition frequency and injury risk, with athletes engaging in more competitions more likely to experience multiple injuries.

## 4. Discussion

This study aimed to investigate the injury patterns, risk factors, and recovery strategies in a cohort of Italian BJJ practitioners. The findings highlighted that BJJ, while offering many physical and mental benefits in adults [1] and children [2], exposes participants to a substantial risk of injury, particularly during training sessions.

The most injured area, as observed in this study, was the knee joint, followed by the shoulder and elbow. The prevalence of knee injuries is in line with previous research on BJJ [3,10,13,28] and other combat sports like Judo [6] and Wrestling [9], where the lower limbs are frequently subjected to high levels of stress due to techniques that involve leg sweeps, takedowns, and guard passes.

In Hasegawa et al. [29], the most injured anatomical district in BJJ injuries presenting to U.S. emergency departments is the upper trunk and shoulder. This data is also consistent with the findings of our survey: in fact, considering only injuries that required an emergency room visit, there is a greater involvement of the upper limbs and upper trunk (n = 31), compared to those involving the lower limbs (n = 26).

In particular, the technique of “passing guard” was identified as a frequent cause of injury, supporting findings from previous studies that emphasize the biomechanical strain that this movement imposes on the knees and other joints [28,30]. It is also important to highlight how the study by Scoggin et al. [12], which analyzed injury mechanisms in BJJ competitions, shows the vulnerability of the elbow, particularly because of the “arm bar”. The “arm bar” in BJJ is a common submission technique that targets the opponent’s elbow joint. It involves trapping one of the opponent’s arms, controlling their wrist, and using legs and hips to hyperextend the elbow by applying pressure in the opposite direction of its natural movement. This maneuver can expose the elbow to potential injuries.

Interestingly, most injuries occurred during training (n = 291; 87.9%) rather than competition (n = 40; 12.9%), in line with other studies on BJJ [12,13,17]. This could be due to several elements, including the fact that training involves the repetitive practice of techniques and longer duration compared to competition bouts. Moreover, training partners in BJJ may vary greatly in skill level, potentially increasing the likelihood of injury due to uneven matchups or incorrect technique execution. This observation suggests that injury prevention strategies should focus not only on improving technique but also on ensuring proper training intensity and partner pairing during practice [12].

This study also found that athletes with advanced belts (i.e., higher skill levels) reported a higher number of injuries, particularly in competition. While it is often assumed that experience should lead to a reduction in injury risk, this finding suggests that the intensity of competition and the physical demands placed on advanced practitioners may offset the protective effect of experience [13]. More experienced athletes may engage in more aggressive techniques, leading to increased exposure to injury risks. These results align with prior studies on BJJ athletes, where higher-level practitioners often experience more injuries due to the intensity and frequency of their participation in competitions [14,28].

The Random Forest Algorithm to rank the feature importance used in this study provided novel insights into the predictors of injury. The identification of BMI and age as significant predictors suggests that physical attributes play a crucial role in injury risk, particularly during training sessions. Higher BMI may increase the load on joints during movements, particularly in grappling sports like BJJ, where athletes must support both their own body weight and that of their opponent. Age-related changes in flexibility and joint mobility may also contribute to the increased risk of injury among older practitioners [31]. This underscores the importance of tailored training programs that consider an athlete’s age and body composition to reduce the risk of injury.

From a rehabilitation perspective, most athletes in this study utilized physiotherapy and therapeutic exercises for recovery, which aligns with best practices in sports medicine [32,33].

Interestingly, a significant number of injured athletes mentioned that they did not seek treatment, possibly indicating a cultural mindset within the BJJ community of ‘pushing through’ injuries or a lack of understanding about the importance of adequate rehabilitation. This approach can be concerning, as neglecting injuries may result in long-term problems and extended recovery periods [10,17]. Coaches and physiotherapists working with BJJ athletes should emphasize the importance of following appropriate treatment protocols after an injury to ensure full recovery and prevent long-term complications.

A particularly valuable addition to this study was the assessment of Psychological Readiness using the I-PRRS scale. Psychological readiness to RTS is an often-overlooked aspect of recovery, yet it may play a role in an athlete’s successful return to competitive performance [19,34].

Athletes who returned to sport before feeling mentally prepared were more likely to experience performance anxiety or fear of reinjury, which may hinder their ability to perform at their pre-injury level [35]. The moderate confidence levels reported by athletes in this study suggest that psychological factors should be addressed as part of the rehabilitation process, alongside physical recovery. This finding agrees with recent research highlighting the importance of psychological support in the rehabilitation process for injured athletes [36].

In addition to these findings, this study sheds light on the broader issue of injury prevention in BJJ. The high injury rate observed during training sessions suggests that coaches and practitioners need to implement more structured injury prevention programs [14,28]. Strengthening exercises targeting the knee, shoulder, and elbow joints, as well as education on proper technique execution, should be emphasized. Furthermore, strategies such as the periodization of training intensity and improved warm-up protocols could help reduce the incidence of injury, particularly for older athletes and those with higher BMI [37,38].

Although the results provide a significant overview of injuries and rehabilitation methods among BJJ practitioners in Italy, this study has some limitations. Firstly, the sample used, while representative, is composed of athletes who voluntarily participated in an online questionnaire, which may introduce a selection bias. The data collected are based solely on self-reported responses from athletes, without independent clinical verification, which could lead to recall errors or distortions in the perception of injury and recovery.

Furthermore, this study did not consider potential variations in training intensity or injury prevention strategies adopted by different clubs, which could influence the incidence of injuries. Despite the use of the Random Forest algorithm, which provided valuable insights into the predictive factors of injury, the predictive analysis remains limited by the lack of prospective data. Future studies should consider the long-term monitoring of athletes to analyze injury risk and recovery times in greater detail. A further limitation of this study is related to the validation process of the survey instrument. Although we performed a content validation through expert review (three coaches and three physiotherapists), a complete psychometric evaluation was not carried out. Additionally, this study focused exclusively on Italian BJJ athletes, which limits the generalizability of the findings to other populations. However, this focus was deliberate, aiming to provide a specific overview of injury patterns and rehabilitation practices within the Italian BJJ community, which has received limited attention in the scientific literature.

Finally, the findings from this study have important implications for the medical and sporting communities. Orthopedic surgeons and physiotherapists should consider the unique demands of BJJ when treating injured athletes, ensuring that rehabilitation programs are tailored to the specific movements and stresses involved in the sport. Likewise, trainers and athletes must be educated on the importance of injury prevention, early treatment, and addressing both physical and psychological readiness before returning to competition [23,39]. Future developments could also include the implementation of a machine learning approach for injury risk prediction as a tool for prevention.

## 5. Conclusions

This study fills a gap in the literature regarding injury incidence and rehabilitation practices in BJJ in Italy. The findings highlight the knee as the most injury-prone joint, followed by the shoulder and elbow, with most injuries occurring during training rather than competition. While most athletes sought physiotherapy for recovery, a significant portion reported not seeking any treatment, raising concerns about potential long-term complications from untreated injuries.

The predictive analysis identified BMI and age as critical factors in injury risk, suggesting that personalized training programs based on these variables could help reduce injury rates. Additionally, the moderate scores on the I-PRRS scale emphasize the importance of psychological readiness for a safe and effective RTS, indicating that rehabilitation should address both physical and mental aspects.

These results carry important implications for coaches, physiotherapists, and athletes, underscoring the need to implement injury prevention strategies and raise awareness of the importance of comprehensive rehabilitation. Future research should incorporate larger samples and longitudinal monitoring to validate these findings and refine clinical guidelines for BJJ while also exploring machine learning approaches for injury risk prediction as a preventive tool.

## Figures and Tables

**Figure 1 jfmk-10-00286-f001:**
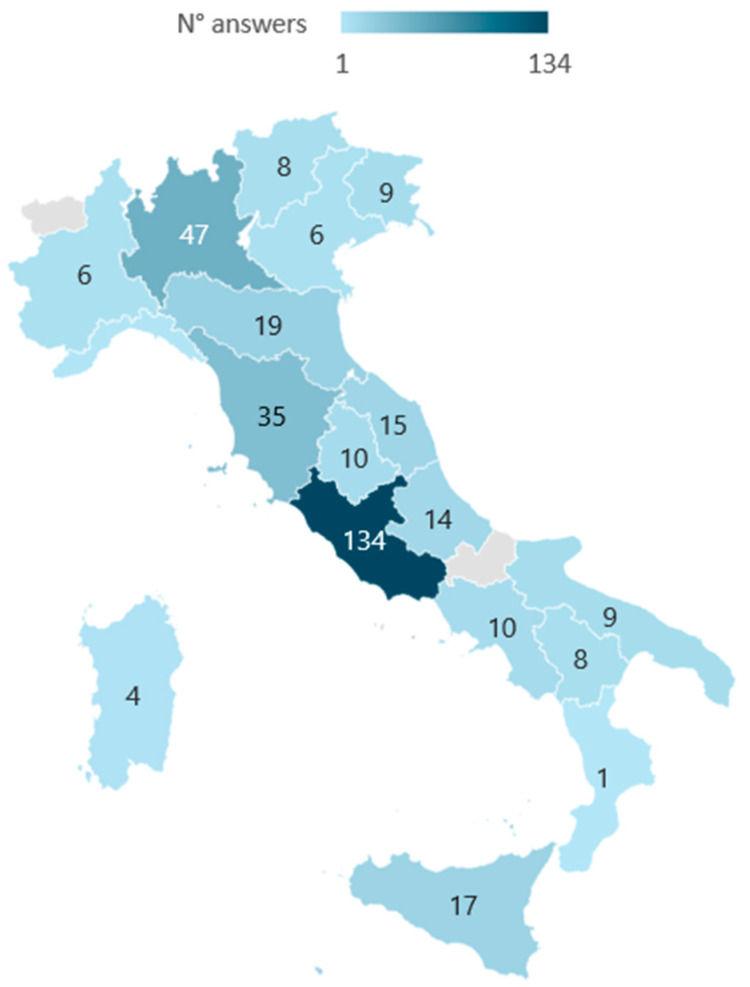
Sample geographical distribution.

**Figure 2 jfmk-10-00286-f002:**
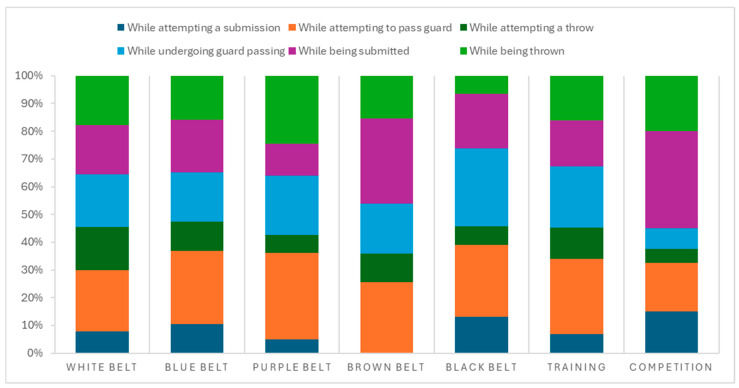
Traumatic mechanisms of injuries in training or competition and belt ranking.

**Figure 3 jfmk-10-00286-f003:**
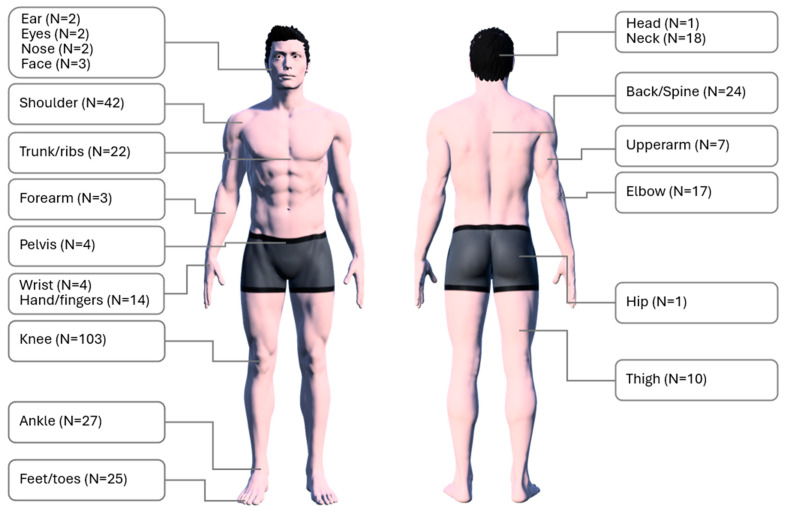
Anatomical distribution of injuries (n = 331); the image is created using the MagicPoser app (version 1.49.1). Available at https://magicposer.com.

**Figure 4 jfmk-10-00286-f004:**
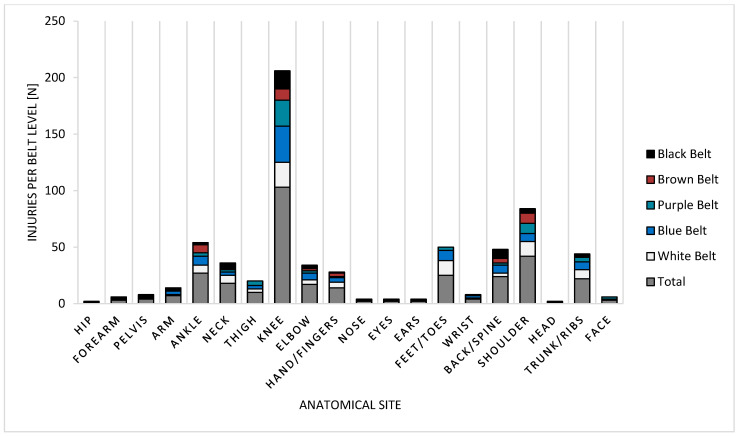
Relationship between injuries (n = 331), anatomical sites, and belt ranking.

**Figure 5 jfmk-10-00286-f005:**
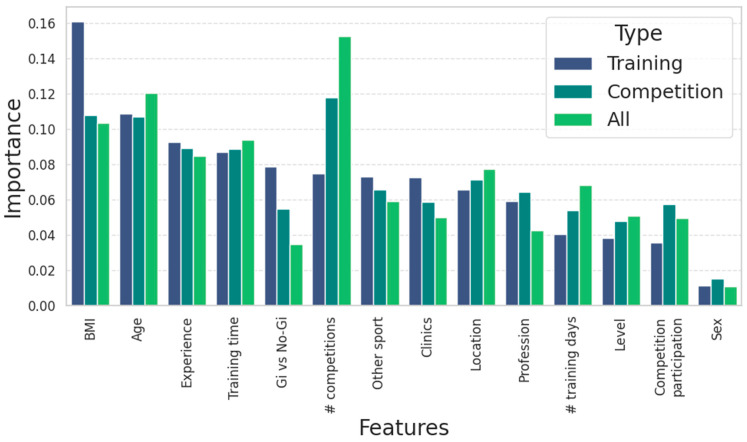
Feature importance in predicting injury occurrence in Brazilian Jiu-Jitsu athletes, categorized by training, competition, and overall contexts. The ‘#’ symbol in the feature names indicates the number of occurrences.

**Figure 6 jfmk-10-00286-f006:**
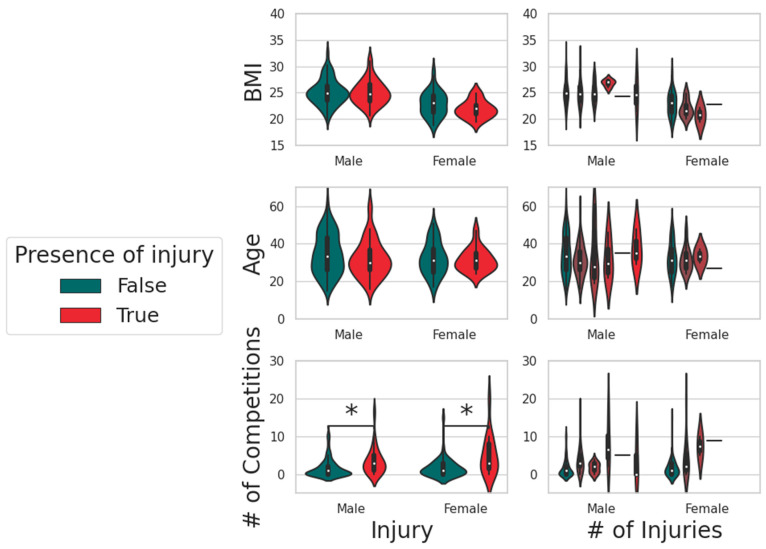
Distribution of BMI, age, and number of competitions by injury status. Violin plots show the distribution of key variables among male and female athletes. Statistically significant differences are marked by asterisks.

**Table 1 jfmk-10-00286-t001:** Sample demographic characteristics.

Characteristics	Participants (n = 360)
Age (years), means ± st. dev.	33.7 ± 10.08
Gender (male), % (n)	83.3 % (300)
Weight (Kg), means ± st. dev.	76.4 ± 11.6
<57.5 Kg, % (n)	6.1% (22)
57.6–64 Kg, % (n)	8.8% (32)
64.1–70 Kg, % (n)	16.4% (59)
70.1–76 Kg, % (n)	19.2% (69)
76.1–82.3 Kg, % (n)	20.8% (75)
82.4–88.3 Kg, % (n)	13.4% (50)
88.4–94.3 Kg, % (n)	7.8% (28)
94.4–100.5 Kg, % (n)	5% (18)
>100.5 Kg, % (n)	1.9% (7)
Height (cm), means ± st. dev.	175.4 ± 8.06
BMI means ± st. dev.	24.7 ± 2.6

**Table 2 jfmk-10-00286-t002:** Sample employment characteristics.

Occupation	Participants (n = 360)
Student, % (n)	33.7 ± 10.08
Manager, % (n)	83.3% (300)
Business, Finance, and Administration, % (n)	76.4 ± 11.6
Health Professional, % (n)	6.1% (22)
Trade and Transport, % (n)	8.8% (32)
Manufacturing and Public Utilities, % (n)	16.4% (59)
Art, Culture, Entertainment, and Sports, % (n)	19.2% (69)
Education, Law, Social Services, Community, and Government Services, % (n)	20.8% (75)
Natural and Applied Sciences and Related Professions, % (n)	13.4% (50)
Natural Resources, Agriculture, and Related Production Occupations, % (n)	7.8% (28)
Student, % (n)	5% (18)
Manager, % (n)	1.9% (7)
Business, Finance, and Administration, % (n)	175.4 ± 8.06
Health Professional, % (n)	24.7 ± 2.6

**Table 3 jfmk-10-00286-t003:** Belt grades, competing and non-competing athletes.

Belt Grades	n	%	Non-Competing Athletes (n)	Competing Athletes (n)
White Belt	100	27.80%	66	44
Blue Belt	105	29.20%	33	72
Purple Belt	67	18.60%	16	51
Brown Belt	41	11.30%	6	35
Black Belt	47	13.10%	13	34
White Belt	100	27.80%	66	44

**Table 4 jfmk-10-00286-t004:** Gi and No-Gi frequencies and percentage in training.

Gi and No-Gi Training	(n)	(%)
100% Gi	43	11.90%
75% Gi and 25% No-Gi	185	51.40%
50% Gi and 50% No-Gi	86	23.90%
25% Gi and 75% No-Gi	25	6.90%
100% No-Gi	21	5.90%
Total	360	100.00%

**Table 5 jfmk-10-00286-t005:** Data related to the number and percentage of athletes based on the belt rank achieved and the number of injuries during training and competitions based on the belt rank achieved.

Belt Grades	n of Athletes	Training		Competition	
		Total Injuries	Injuries Per Athletes	Total Injuries	Injuries Per Athletes
White	100	201	2.01	19	0.19
Blue	105	231	2.2	27	0.25
Purple	67	143	2.13	26	0.39
Brown	41	101	2.46	20	0.49
Black	47	102	2.17	22	0.47

**Table 6 jfmk-10-00286-t006:** Return to play time for BJJ.

Time to Return to Play After Injury	n
Within 1 month	217
Within 3 months	69
Within 6 months	17
Within 12 months	7
More than 12 months	3
I am currently injured and have not resumed sporting activity yet	18
Total	331

**Table 7 jfmk-10-00286-t007:** Injury-Psychological Readiness to Return to Sport scale (I-PRRS) values.

Items	Value
My overall confidence to play is	72.28
My confidence to play without pain is	64.55
My confidence to give 100% effort is	82.31
My confidence in injured body part to handle the demands of the situation is	71.70
My confidence in skill level/ability is	75.55
My confidence to not concentrate on the injury	75.76

## Data Availability

The complete dataset generated during the current study is available from the corresponding author upon reasonable request.

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
