# Peer review of "Is Brazilian Jiu-Jitsu a Traumatic Sport? Survey on Italian Athletes’ Rehabilitation and Return to Sport"

_jfmk, 2025, doi:10.3390/jfmk10030286_

Round 1

Reviewer 1 Report

Comments and Suggestions for Authors

This study adds value to our current understanding of injury patterns in BJJ. The paper is well-organized, and the authors have provided sufficient details throughout the manuscript. This study has a gap in addressing the limited data available on BJJ injuries. The authors have reviewed appropriate published data on BJJ injuries for comparison with their own data, which shows some differences in injury patterns and epidemiology based on the level of training and the study setting. The findings of the current study in terms of key injury-modulating factors (BMI, belt level, and age) are consistent with those of previous studies.

The Introduction section is well written. This study aimed to identify the types of injuries and the most frequent traumatic mechanisms of injuries, as well as the commonly employed treatment and rehabilitation methods. The study aims and hypotheses are well described. The Materials and Methods section is comprehensive and includes a detailed description of the study design, participants, inclusion and exclusion criteria, and method of analysis. The authors stated that since no specific standardized instrument was available for the purpose of the study, they developed their own questionnaire. The authors stated that this new instrument was validated by three coaches and three physiotherapists. Is this sufficient validation? In the participant section, the authors stated that all Italian BJJ athletes were included. Why is it limited to Italian nationalities? The Results section is well organized, and the tabulated data is easy to understand. In addition, the Figures also add to the ease of understanding and interpreting the data. The Discussion includes a comparison with other published studies. The authors have appropriately detailed the limitations of their study. Of particular importance is the authors’ description of the practical implications of the findings of their study in the recognition, treatment, rehabilitation, and prevention of injuries in BJJ. These Conclusions were supported by the study data.

Author Response

Reviewer 1

This study adds value to our current understanding of injury patterns in BJJ. The paper is well-organized, and the authors have provided sufficient details throughout the manuscript. This study has a gap in addressing the limited data available on BJJ injuries. The authors have reviewed appropriate published data on BJJ injuries for comparison with their own data, which shows some differences in injury patterns and epidemiology based on the level of training and the study setting. The findings of the current study in terms of key injury-modulating factors (BMI, belt level, and age) are consistent with those of previous studies.

The Introduction section is well written. This study aimed to identify the types of injuries and the most frequent traumatic mechanisms of injuries, as well as the commonly employed treatment and rehabilitation methods. The study aims and hypotheses are well described. The Materials and Methods section is comprehensive and includes a detailed description of the study design, participants, inclusion and exclusion criteria, and method of analysis. The authors stated that since no specific standardized instrument was available for the purpose of the study, they developed their own questionnaire. The authors stated that this new instrument was validated by three coaches and three physiotherapists. Is this sufficient validation? In the participant section, the authors stated that all Italian BJJ athletes were included. Why is it limited to Italian nationalities? The Results section is well organized, and the tabulated data is easy to understand. In addition, the Figures also add to the ease of understanding and interpreting the data. The Discussion includes a comparison with other published studies. The authors have appropriately detailed the limitations of their study. Of particular importance is the authors’ description of the practical implications of the findings of their study in the recognition, treatment, rehabilitation, and prevention of injuries in BJJ. These Conclusions were supported by the study data.

Authors’ response to reviewer 1:

Thank you for your thoughtful comment. We appreciate the opportunity to clarify the aspects related to this comment “The authors stated that this new instrument was validated by three coaches and three physiotherapists. Is this sufficient validation? In the participant section, the authors stated that all Italian BJJ athletes were included. Why is it limited to Italian nationalities?”

On questionnaire validation: the questionnaire we used was adapted from a previously published and validated survey by Petrisor et al. (2019), to which we added items to better capture rehabilitation processes and psychological readiness in BJJ athletes. The adapted version was subjected to content validation through expert review involving three BJJ coaches and three physiotherapists experienced in combat sports. This step aimed to ensure content relevance, clarity, and cultural appropriateness for the Italian BJJ context. While we acknowledge that a more extensive validation (e.g., involving psychometric testing or Delphi methodology) would strengthen the tool, our primary objective was to ensure that the questionnaire was understandable and covered the key dimensions relevant to injury and rehabilitation in this population. We have clarified this point in the manuscript and explicitly referred to this process as a pretest for face and content validity, rather than a full psychometric validation.

On participant nationality: the study focused on Italian BJJ athletes because the aim was to provide a snapshot of injury patterns, rehabilitation practices, and RTS in the Italian BJJ community, which has been growing rapidly in recent years but remains underrepresented in the literature. By limiting inclusion to athletes of Italian nationality or those training in Italy and fluent in Italian, we ensured the linguistic and cultural consistency required for reliable interpretation of questionnaire responses, particularly for items related to psychological readiness.

Reviewer 2 Report

Comments and Suggestions for Authors

Thank you for the opportunity to review this interesting manuscript. Overall, is well structured and written. Please see below some comment ordered by sections and lines.

INTRODUCTION

Line 43, explode the acronym BJJ as appear first in the main manuscript

Line 56, please add a reference 

Line 78, explode the acronym RTS as appear first in the main manuscript

METHODS

line 90, use the term "conducted" instead of "carried out"

Line 109, add these reference after the "."
https://pubmed.ncbi.nlm.nih.gov/33434869/

line 112-114, please report the time period in which the survey was available to participants 

line 113, this should be reported in the results section.

line 143-144, you mentioned "studies" but used a single reference. 

line 176-177, this should be reported in the results section.

Line 182, delete the typo ")"

RESULTS

line 249, probably you meant "remaining"

Line 251, delete "in general"

Line 258, please replace the sentence ", as shown in Figure 3," with "(Figure 3)"

Line 262, please replace the sentence ", as shown in Figure 4," with "(Figure 4)"

Line 270-271, please re-phrase the following "stated that their injury was of a nature classified as "Other" compared to the listed options."

Line 303 & 305 & 324, please double check "Error! Reference source not found." It seems that the reference software did not reported the reference.

Line 308 & 385, this sentence is to definitive for the study design "plays a crucial". I suggest changing in "may play a role"

Line 315, "shown as # competitions", where?

GENERAL
I can't see the STROBE checklist attached. Please submit it if not already present and cite in within the manuscript in the Methods section.

Author Response

Reviewer 2

Thank you for the opportunity to review this interesting manuscript. Overall, is well structured and written. Please see below some comment ordered by sections and lines.

Authors’ response to the reviewer 2:

We would like to thank the reviewer for the positive feedback and constructive suggestions. Below, we provide point-by-point responses to each comment. In bold our responses. In the text you will find the corrections highlighted.

INTRODUCTION

Line 43, explode the acronym BJJ as appear first in the main manuscript

Thank you for the comment, we corrected.

Line 56, please add a reference 

We added the references to show how judo and wrestling have lots of published data.

Line 78, explode the acronym RTS as appear first in the main manuscript

 Thank you for the comment, we corrected.

METHODS

line 90, use the term "conducted" instead of "carried out"

 Thank you for the comment, we corrected.

Line 109, add these reference after the "."
https://pubmed.ncbi.nlm.nih.gov/33434869/
Thank you for the comment, we added the reference as suggested.

line 112-114, please report the time period in which the survey was available to participants 

Thank you, we added as requested.

line 113, this should be reported in the results section.

Thank you, we moved as requested.

line 143-144, you mentioned "studies" but used a single reference. 

 Thank you for the comment, we corrected.

line 176-177, this should be reported in the results section.

Thank you, we removed this sentence.

Line 182, delete the typo ")"

 Thank you for the comment, we corrected.

RESULTS

line 249, probably you meant "remaining"

 Thank you for the comment, we corrected.

Line 251, delete "in general"

 Thank you for the comment, we corrected.

Line 258, please replace the sentence ", as shown in Figure 3," with "(Figure 3)"

Line 262, please replace the sentence ", as shown in Figure 4," with "(Figure 4)"

 Thank you for the comment, we corrected.

Line 270-271, please re-phrase the following "stated that their injury was of a nature classified as "Other" compared to the listed options."

Thank you we rephrased: “…reported sustaining an injury that did not fall under the predefined categories and was therefore classified as 'Other'….

Line 303 & 305 & 324, please double check "Error! Reference source not found." It seems that the reference software did not reported the reference.

Thank you we corrected, it was an error to the linked figure.

Line 308 & 385, this sentence is to definitive for the study design "plays a crucial". I suggest changing in "may play a role"

 Thank you for the comment, we corrected.

Line 315, "shown as # competitions", where?

Thank you we added “in figure 6”.

GENERAL
I can't see the STROBE checklist attached. Please submit it if not already present and cite in within the manuscript in the Methods section.

 Thank you for the comment, we added supplementary material.
